

# Towards traversable wormholes from force-free plasmas

**Nabil Iqbal⋆ and Simon F. Ross†**

Centre for Particle Theory, Department of Mathematical Sciences
Durham University, South Road, Durham DH1 3LE, UK

⋆ nabil.iqbal@durham.ac.uk, † s.f.ross@durham.ac.uk

## Abstract

The near-horizon region of magnetically charged black holes can have very strong magnetic fields. A useful low-energy effective theory for fluctuations of the fields, coupled to electrically charged particles, is force-free electrodynamics. The low energy collective excitations include a large number of Alfven wave modes, which have a massless dispersion relation along the field worldlines. We attempt to construct traversable wormhole solutions using the negative Casimir energy of the Alfven wave modes, analogously to the recent construction using charged massless fermions. The behaviour of massless scalars in the near-horizon region implies that the size of the wormholes is strongly restricted and cannot be made large, even though the force free description is valid in a larger regime.


# 1 Introduction

Near-extremal charged black holes have many interesting features. Recently a new one was pointed out: in [1,2], it was observed that for an extensive range of values of the total magnetic charge, the near-horizon region of a magnetically charged black hole has strong magnetic fields. If we consider electrically charged particles moving in the near-horizon region, the Landau levels of the charges moving along the strong magnetic fields give a large number of light fields in the two-dimensional space along the field lines. In [1], it was argued that the Casimir energy from these fields could lead to traversable wormhole geometries in a single universe. [2] further explored the effects of these fields on radiation from the charged black hole.

Those works used a microscopic treatment of charged particles interacting with electromagnetic fields; however, coarse-grained effective descriptions of this regime exist. In this paper, we explore the use of the effective theory of force-free electrodynamics (FFE) to describe field fluctuations in the near-horizon region of magnetically charged black holes. As we will briefly review below, FFE can be thought of as a hydrodynamic theory describing strongly magnetized plasmas in a regime where the temperature may be neglected. A particularly interesting fact is that this theory contains a large number of light collective modes, called Alfven wave modes; these may be thought of as transverse oscillations of magnetic field lines, and in our model will behave as light scalar fields in the two-dimensional space along the field lines.

One would hope that we could use the Casimir energy of the Alfven wave modes to construct traversable wormholes, in analogy with the construction of [1]. Traversable wormholes have been studied intensively in the last few years. They are interesting as theoretical objects, offering a new insight into the role of entanglement in constructing spacetime geometry in holographic theories [3–5]. Given that it is possible to make traversable wormholes in theory, it is obviously interesting to see if they could actually exist in the real universe. The initial construction of [1] provides a mechanism for the existence of self-supporting traversable wormholes in a single universe, but the wormholes realised there are quite small; essentially they are required to be smaller than the length scale set by the mass of the charged particle in question (i.e. the electron mass).

In this work we attempt to build larger wormholes using the low energy field content of our universe. The main novelty is that the light fields we consider are not microscopic degrees of freedom but rather collective Alfven modes. Naively, it would appear that using them as the energy source could provide such a construction for larger wormholes; the lightness of these collective modes comes from general principles of effective theory, and we are no longer constrained by the microscopic length scale set by the electron. Indeed, the FFE description is valid for black holes with horizon radii up to $10^7$ m. (See also [6] for a different approach to constructing larger wormholes, and [7–13] for other work on self-supporting wormholes.)

Unfortunately for our ambitions at interstellar construction, this proves difficult to implement in detail. Firstly, the answers depend sensitively on the UV completion to the effective theory of FFE. We compute corrections to the relevant dispersion relation to the Alfven waves subject to some conservative assumptions. Secondly, the different scaling dimensions for massless scalars and fermions in the near-horizon region also imply significant differences in their effects. This is seen most clearly if we consider just this near-horizon $AdS_2$ region. We can introduce an explicit coupling between the modes on the two boundaries of $AdS_2$. In [14], the theory with such a coupling for light fermions was found to have an eternal wormhole solution. But if we consider massless scalars instead, the potential for the length of the wormhole does not have a minimum.

Following [1], we then look for wormhole solutions with two oppositely charged mouths in an asymptotically flat space. The results from the pure $AdS_2$ analysis suggests that con-

structing wormholes from the Casimir energy of light scalars may be more difficult than in the fermionic case. We indeed find that the potential for the length of the wormhole does not have a minimum, in the regime where the wormhole is long enough that the effects of the AdS$_2$ region dominates. However, we can find a minimum at short wormhole lengths, where the effects of the asymptotically flat region becomes relevant. We can thus construct wormhole solutions within the FFE effective theory. Unfortunately, these wormholes are small; the maximum throat radius is of order $10^{-21}$ metres. The effective theory actually breaks down before we get to such small scales, where we would need to take into account contributions of additional standard model fields, as in [2].

In the next section, we will review the general theory of FFE and discuss the appearance of the Alfven wave excitations, and the limitations of the effective field theory. In Section 3, we set up the application to the near-extremal magnetically charged black holes, reviewing the geometry and focusing on the near-horizon AdS$_2$ region. In Section 4, we consider the pure AdS$_2$ geometry, introducing a boundary coupling for the operators dual to the Alfven waves, and find that the Casimir energy for this boundary condition scales as $1/\ell^{2\Delta}$, where $\Delta$ is the dimension of the boundary operator dual to the bulk field. This produces the essential difference between massless fermions, with $\Delta = \frac{1}{2}$, and massless scalars, with $\Delta = 1$. In Section 5, we consider the Casimir energy for massive fields with periodic boundary conditions on AdS$_2$, and find that it has the same scaling, except for fields sufficiently close to zero mass, where it scales as $1/\ell$. Finally in Section 6, we see that this restricts the number of Alfven wave modes that contribute to the wormhole construction, giving a total Casimir energy which scales as $1/\ell^2$ for large $\ell$. There is a cross-over to $1/\ell$ scaling at small $\ell$, allowing for the construction of a traversable wormhole, but only for small wormholes which lie outside the range of validity of our discussion.

## 2 Force-free electrodynamics

Force-free electrodynamics is an effective theory describing Maxwell electrodynamics coupled to electrically charged matter, in a state where there is an extremely strong magnetic field [15–17]. It is usually considered in situations with a plasma of charged particles which effectively screens the component of the electric field along the magnetic field, for example in astrophysics where it is thought to describe the magnetospheres of pulsars [18,19]. The equations of motion of the theory are

$$F_{\sigma\nu}\nabla_\mu F^{\mu\nu} = 0, \qquad \nabla_{[\mu}F_{\rho\sigma]} = 0, \qquad \epsilon^{\mu\nu\rho\sigma}F_{\mu\nu}F_{\rho\sigma} = 0. \tag{1}$$

We will also use differential forms notation, in which the latter two equations can be rewritten as $dF = 0$, $F \wedge F = 0$. The first equation is the force-free condition; the coupling of the electromagnetic field to the current sourcing the field vanishes. The second is the usual Bianchi identity, and the last is the Lorentz-invariant notion of the electric field being screened to zero in the direction of the magnetic field, that is $\vec{E} \cdot \vec{B} = 0$. An electromagnetic field satisfying this last condition is said to be degenerate. A review of the traditional formulation of this theory can be found in [17]; it can be thought of as a form of magnetohydrodynamics which is at "zero temperature", in that there is no preferred rest frame. In its conventional formulation, it is usually understood that the stress-energy of the electromagnetic field is much higher than that of the charged matter screening the electric field, which can thus be neglected.

This theory has recently attracted attention in the context of higher-form symmetries. A connection with higher-form symmetries was first made in [20]. It was recently reformulated as an effective field theory in [21], and further work on the higher-form symmetry viewpoint can be found in [22–25].

The theory is often considered in situations with a plasma density, but the theory is still useful for describing fluctuations around vacuum electromagnetic backgrounds satisfying the degeneracy condition $F \wedge F = 0$, if the background magnetic field is strong enough; in response to fluctuations, charges can be easily pair-produced, screening the electric field to zero over long distance scales. We will consider the theory in this setting.

## 2.1 Alfven waves

A convenient formulation of FFE was introduced in [26]. Introduce a Lagrange multiplier $\Phi$ enforcing the constraint that $F \wedge F = 0$, and write the action

$$S = \int d^4x \sqrt{-g}\left(-\frac{1}{4g^2}F^2 - \frac{1}{32\pi^2}\Phi\epsilon^{\mu\nu\rho\sigma}F_{\mu\nu}F_{\rho\sigma}\right) = \int \left(-\frac{1}{4g^2}F \wedge \star F - \frac{1}{32\pi^2}\Phi F \wedge F\right), \quad (2)$$

where the field strength $F$ is the derivative of a vector potential as in ordinary electrodynamics, $F = dA$, so the fundamental fields in this formulation are $A_\mu$ and $\Phi$. Then the Bianchi identity $dF = 0$ is trivially satisfied. The equations of motions from this action are

$$\nabla_\nu F^{\mu\nu} = -\frac{g^2}{8\pi^2}\epsilon^{\mu\nu\rho\sigma}\nabla_\nu\Phi F_{\rho\sigma}, \quad \epsilon^{\mu\nu\rho\sigma}F_{\mu\nu}F_{\rho\sigma} = 0, \quad (3)$$

or in differential forms notation

$$d \star F = -\frac{g^2}{8\pi^2}d\Phi \wedge F, \quad F \wedge F = 0. \quad (4)$$

The second equation implies that $F = \alpha \wedge \beta$ for some one-forms $\alpha, \beta$. Contracting the first equation with $F_{\mu\gamma}$ and using this decomposition then implies the other FFE equation, $F_{\mu\gamma}\nabla_\nu F^{\mu\nu} = 0$.

This formulation has two advantages; it gives a straightforward description of the Alfven wave mode, and Gralla has described a derivation of it in strong fields which includes a correction to this effective field theory description [27]. To see the first, consider the linearization of these equations about some degenerate background field $F^0$ with $d \star F^0 = 0$, so $F = F^0 + f$, $\Phi = \phi$ in terms of the linear perturbations $f = da$, $\phi$. The linearized equations of motion are

$$d \star f = -\frac{g^2}{8\pi^2}d\phi \wedge F^0, \quad F^0 \wedge f = 0. \quad (5)$$

We will consider cases where the background field only has components in two of the directions, so we split the coordinates $x^\mu$, $\mu = 0, \ldots 3$ into two subspaces $x^a$ and $x^i$, $i = 0, 1$, $a = 2, 3$, and assume the non-zero components of $F^0$ are $F^0_{ab} = B\epsilon_{ab}$. We also assume the metric has a product structure, so

$$ds^2 = g_{ij}(x^i)dx^i dx^j + g_{ab}(x^a)dx^a dx^b. \quad (6)$$

This ansatz encompasses a uniform magnetic field in flat space (where $g_{ij}$ and $g_{ab}$ are flat metrics), as well as the near-horizon $AdS_2 \times S^2$ geometry of the black hole solutions we will consider. The second equation of motion then implies that $f_{ij} = 0$, and the non-trivial components of the first equation are

$$(d \star f)_{ija} = 0, \quad (d \star f)_{iab} = -\frac{g^2}{8\pi^2}\partial_i\phi B\epsilon_{ab}. \quad (7)$$

Contracting the latter with $\epsilon^{ab}$ gives $\epsilon_{ij}\nabla_c f^{jc} = -\frac{g^2}{8\pi^2}B\partial_i\phi$, where $\nabla_c$ is the covariant derivative with respect to $g_{ab}$. Taking a derivative, we have

$$\begin{aligned}
\nabla_i\nabla^i\phi &= -\frac{8\pi^2}{g^2 B}\frac{1}{\sqrt{-g}}\partial_i(\sqrt{-g}\epsilon^{ij}\nabla_c f^c_j) = -\frac{8\pi^2}{g^2 B}\epsilon^{ij}\partial_i\nabla_c f^c_j \\
&= -\frac{8\pi^2}{g^2 B}\epsilon^{ij}(\partial_i\partial_j\nabla_c a^c - \nabla_c\nabla^c\partial_i a_j) = 0,
\end{aligned} \quad (8)$$

where $\nabla_i$ is the covariant derivative with respect to $g_{ij}$. In the last step the first term vanishes by the symmetry of the derivatives, and the second term vanishes as $f_{ij} = 0$. Thus, the perturbation field $\phi(x^i, x^a)$ satisfies a two-dimensional massless wave equation in the $x^i$ coordinates, independent of the dependence on $x^a$. These are the Alfven wave modes. As they depend only on the $x^i$, they can be thought of as waves propagating *along* the magnetic field lines, where the speed of propagation is independent of their transverse momentum.

In this analysis, we have treated the gauge theory as progagating on a fixed background geometry, but in the next section, we want to consider the FFE effective theory coupled to gravity. The presence of the background field $F^0$ implies that the perturbations will couple nontrivially to metric perturbations at linear order, and we need to solve the coupled equations. Nevertheless, we still obtain a decoupled set of Alfven wave modes. The differential forms representation of the equations of motion makes it easy to see this, as they make it manifest that the dependence on the metric is limited to the Hodge star, which involves the determinant of the metric. Under a perturbation $g_{\mu\nu} = g^0_{\mu\nu} + \delta g_{\mu\nu}$, the determinant $g = g^0(1 - g^0_{\mu\nu}\delta g^{\mu\nu})$, so the linearized equations including a metric perturbation are

$$d \star f - \frac{1}{2}d(g^0_{\mu\nu}\delta g^{\mu\nu}) \wedge F^0 = -\frac{g^2}{8\pi^2}d\phi \wedge F^0, \quad F^0 \wedge f = 0, \tag{9}$$

where the star is with respect to the background metric $g^0$. There will also be a linearised Einstein equation which determines $\delta g_{\mu\nu}$, but we do not write it explicitly as it does not enter into the argument for the Alfven wave mode. With the same assumptions as before that the non-trivial components of $F^0$ are $F^0_{ab} = B\epsilon_{ab}$ and the metric has a product structure, the second equation implies $f_{ij} = 0$ and the non-trivial components of the first are

$$(d \star f)_{ija} = \frac{1}{2}\partial_a(g^0_{\mu\nu}\delta g^{\mu\nu})B\epsilon_{ij}, \quad (d \star f)_{iab} = -\frac{g^2}{8\pi^2}\partial_i\phi B\epsilon_{ab}. \tag{10}$$

The metric perturbation only enters into the first equation, but it is the second that we needed in the argument above, so it goes through as before, and $\phi$ satisfies a two-dimensional wave equation $\nabla_i\nabla^i\phi = 0$.[1]

In this effective field theory description, there are an arbitrary number of two-dimensional massless fields obtained from momentum eigenmodes of the scalar in the transverse space. This divergent density of states is a peculiarity of the IR effective theory. Within this IR effective theory alone, it formally holds to all orders in the derivative expansion (see [20] for an argument to this effect); this seems unphysical, and presumably it is cutoff in a microscopic model. For our later purposes it will be essential to understand how many modes there actually are that contribute in the near-horizon region. We are thus led to examine the validity of force-free electrodynamics.

To our knowledge, while one expects FFE to be valid at strong fields, the *precise* regime of its validity (and how to systematically take into account corrections) is still not very well understood. We will discuss one particular UV completion below, but here we discuss some general principles. One might argue on dimensional grounds that the FFE effective theory is valid only for transverse momenta $k_\perp$ satisfying

$$k_\perp^2 < B. \tag{11}$$

Below we sketch a dynamical argument for this. We can take the perspective that we would expect the FFE description to be good when the stress-energy of the charge carriers is negligible

---

[1]Our discussion will focus on the scalar $\phi$, but note that the Alfven wave fluctuations involve perturbations of the gauge field and metric as well, determined by solving the coupled Maxwell and Einstein equations taking $\phi$ as a source.

compared to the electromagnetic field (following e.g. the arguments of [17]). The stress-energy of the electromagnetic field $T_{\mu\nu}^F \sim g^{-2}B^2$. Suppose the charge carrier was a complex scalar field $\varphi$, with stress-energy

$$T_{\mu\nu}^\varphi = \partial_\mu\varphi^*\partial_\nu\varphi - \frac{1}{2}g_{\mu\nu}\left(|\partial\varphi|^2 - m^2|\varphi|^2\right) \tag{12}$$

and current

$$j_\nu^\varphi = \varphi^*\partial_\nu\varphi - \varphi\partial_\nu\varphi^*. \tag{13}$$

In FFE, the current is related to the magnetic field through $g^2 j = d * F$, so if the fields are varying on a scale $k_\perp$, $j \sim g^{-2}k_\perp B \sim k_\perp|\varphi|^2$. In a relativistic regime, $T_{\mu\nu}^\varphi \sim k_\perp^2|\varphi|^2 \sim g^{-2}k_\perp^2 B$, so $T_{\mu\nu}^\varphi \sim T_{\mu\nu}^F$ at a cutoff scale $k_\perp^2 \sim B$.

The number of Alfven wave modes that we can reliably consider in effective field theory is thus limited by this bound on $k_\perp^2$; in a transverse volume $L^2$, there are $k_\perp^2 L^2 < BL^2$ modes, which is just the total number of units of magnetic flux in the transverse space.

However, in the context of the applications we consider later, this is not the most important limitation on the number of Alfven wave modes. Instead, we get a more important limitation from considering departures from the exact massless dispersion relation in the two-dimensional subspace coming from corrections to the low energy effective theory. We want light fields; if the corrections give the Alfven waves a small mass depending on $k_\perp^2$, then bounding the mass will bound $k_\perp^2$ and hence the number of modes we consider.

## 2.2 Correction to Alfven wave dispersion relation

In principle the Alfven wave dispersion relation will receive corrections arising from the UV completion. Here we discuss one such microscopic model which reduced to FFE in a certain limit. In [27], FFE was derived from an analysis of QED+Maxwell in a strong field coherent plasma regime. The action obtained there included a correction to the above FFE action,

$$S = \int d^4x\sqrt{-g}\left(-\frac{1}{4g^2}F^2 - \frac{1}{32\pi^2}\Phi\epsilon^{\mu\nu\rho\sigma}F_{\mu\nu}F_{\rho\sigma} + \frac{B_0}{8\pi^2}\left(\frac{1}{2}h^{\mu\nu}\nabla_\mu\Phi\nabla_\nu\Phi - m^2(1-\cos\Phi)\right)\right), \tag{14}$$

where $B_0$ is the magnetic field strength and $h^{\mu\nu}$ is a projector along the field sheets. $\Phi$ – which previously in (2) was a Lagrange multiplier enforcing degeneracy of the fields – here arises microscopically as a bosonization of microscopic electrons moving along the magnetic field lines. For degenerate fields, $B_0^2 = \frac{1}{2}F^2$ and $h_{\mu\nu} = B_0^{-2}F_{\mu\alpha}F^\alpha{}_\nu + g_{\mu\nu}$. The mass $m$ is formally a free parameter in this analysis, but it is argued [27] that it should presumably be identified with the mass of the electron. To understand this, note that the winding of $\Phi$ sources the electric charge, and the energy cost of such winding is suppressed by $m^2$, which we can thus associate with the electric charge gap[2].

The correction terms are small compared to the FFE action for strong magnetic fields, as the first two terms scale quadratically with the field, while the other terms only scale linearly. To understand when these corrections are important, let us first note that in a solution to FFE *without* the correction terms, balancing the first two terms in the action leads to $\Phi \sim g^{-2}$. If we assume that derivatives of $\Phi$ are on the same order as the microscopic scale $m$, then we have that the correction is small provided that $B > B_\star$, where[3]

$$B_\star \equiv m^2 g^{-2}. \tag{15}$$

---

[2]This interpretation is somewhat clouded by the very anisotropic treatment of the directions parallel and perpendicular to the magnetic field, but we will simply assume $m$ can be related to the electron mass.

[3]It is worth noting that this is actually much larger than the usual critical field in plasma physics, which in our units is $B_c = m^2$; the different scaling with the dimensionless EM coupling appears to arise from the precise UV completion studied here. Throughout we will make the more conservative assumption that $B \gg B_\star$. It is possible that a different UV completion would allow an extension of the regime of the validity; our work may thus be thought of as the most pessimistic estimate.

We would like to consider the effect of this correction on the dispersion relation for the Alfven wave mode, so we consider again the linearization $F = F^0 + f$, $\Phi = \phi$. As these fluctuations are small, we may replace $\cos\Phi$ with its quadratic approximation. Since the correction term is now quadratic in $\Phi$, the correction will only affect the $\phi$ equation of motion, which becomes

$$\frac{B_0}{8\pi^2}(\nabla_\mu h^{\mu\nu}\nabla_\nu\phi - m^2\phi) + \frac{1}{16\pi^2}\epsilon^{\mu\nu\rho\sigma}F^0_{\mu\nu}f_{\rho\sigma} = 0. \tag{16}$$

With the assumption that $F^0_{ab} = B\epsilon_{ab}$ and the metric has a product structure, we have $B_0 = B$ and $h^{\mu\nu}$ has components only in the first two-dimensional subspace, $h^{ij} = g^{ij}$, so this equation is

$$\frac{B}{8\pi^2}(\nabla_i\nabla^i\phi - m^2\phi) = -\frac{1}{16\pi^2}B\epsilon^{ij}f_{ij}. \tag{17}$$

As before, the $a$ equation of motion implies $\nabla_i\nabla^i\phi = -\frac{8\pi^2}{g^2B}\nabla_c\nabla^c\epsilon^{ij}f_{ij}$, so we have

$$\nabla_i\nabla^i\phi = \frac{16\pi^2}{g^2B}\nabla_c\nabla^c(\nabla_i\nabla^i\phi - m^2\phi). \tag{18}$$

The first term on the RHS is a higher-derivative correction, suppressed by $k_\perp^2/B$, so this is small in the regime $k_\perp^2 \ll B$. The new effect is the second term, which we can regard as an effective mass for the two-dimensional fields,

$$m^2_{\text{eff}} = \frac{16\pi^2}{g^2B}k_\perp^2 m^2 = \frac{16\pi^2 B_\star}{B}k_\perp^2. \tag{19}$$

We see that indeed the Alfven wave mode is approximately massless in the two-dimensional field sheet; the mass is suppressed for strong magnetic fields by $B_\star/B$. Similarly, the stress energy of each Alfven wave mode can be understood as that of an approximately massless collective scalar field moving in the $\text{AdS}_2$ directions; microscopically however this stress energy comes both from the electromagnetic degrees of freedom and from the fermion degrees of freedom bosonized into the field $\Phi$.

Note however that this correction is not a higher derivative effect in the linearised theory; this arises from the fact that FFE is not an IR limit of the theory described by (14), but rather a strong-field limit. In particular, the effective mass in the two-dimensional theory scales as $k_\perp^2$, just as an ordinary Kaluza-Klein mass would. We will see that this mass correction is significant enough in our applications that constraints on $m^2_{\text{eff}}$ will imply stronger bounds on $k_\perp^2$ than the general condition $k_\perp^2 \ll B$. [4]

## 3 Black holes and wormholes

We are interested in applying these FFE ideas to magnetically charged black holes. We will consider a four-dimensional asymptotically flat black hole for definiteness, although some of the ideas will extend naturally to other contexts.

We consider the Reissner-Nordström black hole, with metric

$$ds^2 = -\left(1 - \frac{2GM}{r} + \frac{R^2}{r^2}\right)dt^2 + \left(1 - \frac{2GM}{r} + \frac{R^2}{r^2}\right)^{-1}dr^2 + r^2(d\theta^2 + \sin^2\theta\, d\phi^2) \tag{20}$$

---

[4]It is interesting to note that if we consider modes with $k_\perp^2 \sim B$, then $m_{\text{eff}} \sim m$. It would be interesting to understand better the relation of these Alfven wave modes to the Landau levels of the microscopic QED+Maxwell theory.

and Maxwell field $A = \frac{Q}{2}\cos\theta\, d\phi$, where

$$R^2 = \frac{\pi G Q^2}{g^2}. \tag{21}$$

If we work in conventions where the fundamental unit of electric charge is one, the magnetic charge $Q$ carried by the black hole is integer quantized.

We will be interested in considering near-extremal black holes, which develop a long approximately $\text{AdS}_2 \times S^2$ throat. In the extremal limit $GM = R$, the near-horizon limit $r - R \ll R$ of the charged black hole is an $\text{AdS}_2 \times S^2$ space where both $\text{AdS}_2$ and $S^2$ have radius $R$. Defining $\mu^2 = (GM)^2 - R^2$, and making the coordinate transformation $\tilde{t} = \frac{\mu}{R^2}t$, $r = R + \mu\cosh\rho$, the near-horizon geometry is

$$ds^2 = R^2\left(-\sinh^2\rho\, d\tilde{t}^2 + d\rho^2 + \left(d\theta^2 + \sin^2\theta\, d\phi^2\right)\right). \tag{22}$$

We will later discuss the $\text{AdS}_2$ in the global coordinates

$$ds^2 = \frac{R^2}{\cos^2\sigma}(-d\tau^2 + d\sigma^2), \tag{23}$$

which are related to the near-horizon coordinates by $\sinh\rho\sinh\tilde{t} = \frac{\sin\tau}{\cos\sigma}$, $\cosh\rho = \frac{\cos\tau}{\cos\sigma}$. In these coordinates the $\text{AdS}_2$ boundaries are at $\sigma = \pm\frac{\pi}{2}$.

For a substantial range of values of $Q$ the magnetic field in this near-horizon region is strong. The Maxwell field strength is $F = \frac{Q}{2}\sin\theta\, d\theta\, d\phi$. To determine the strength of the field, it is more appropriate to rewrite this in terms of the proper volume form on the sphere;

$$F = \frac{Q}{2R^2}\epsilon_{S^2}, \tag{24}$$

so the magnetic field is

$$B = \frac{Q}{2R^2} = \frac{g^2}{2\pi G Q}, \tag{25}$$

and the strength of the field is inversely proportional to the charge. Reducing the charge reduces the total amount of flux through the throat, but it shrinks the size of the throat more quickly, and hence the local field density is increasing.

The black hole is a solution of Einstein gravity coupled to a Maxwell field, but the magnetically charged black holes also satisfy the FFE equations of motion, and can be thought of as solutions of FFE coupled to gravity. As mentioned in the previous section, FFE is usually thought of as a theory of plasmas, but it includes as solutions any degenerate vacuum Maxwell field, and an FFE description is useful if the field is strong enough that fluctuations about this background that would produce electric fields violating the FFE equation are efficiently screened by charges produced by vacuum fluctuations; in this case we expect the low-energy fluctuations to be collective plasma modes (such as Alfven waves) rather than free photon excitations.

The strong field condition is $B > B_\star$, where the critical field strength $B_\star = g^{-2}m^2$. The field in the $\text{AdS}_2 \times S^2$ solution satisfies $B > B_\star$ if $Q < Q_c = \frac{g^4}{2\pi G m^2}$. Putting in the mass of the electron and the strength of the real-life couplings, the critical magnetic charge is $Q_c \sim 10^{39}$ times the minimum Dirac monopole. At this maximum charge, the size of the throat is $R \sim 10^7$ m, so this strong field regime includes quite large black holes.

In the naive FFE theory, the Alfven wave modes give exactly massless two-dimensional fields on the $\text{AdS}_2$ spacetime. Expanding in spherical harmonics, $\phi = \sum \phi^{lm}(x^i)Y_{lm}(\theta, \phi)$,

the $\phi^{lm}$ are massless scalars on AdS$_2$. Taking into account the correction term in (14), these fields get a small mass. Plugging $k_\perp^2 = \frac{l(l+1)}{R^2}$ into (19) we have

$$m_{\text{eff}}^2 = \frac{32\pi^2}{g^2 Q} l(l+1) m^2 \,. \tag{26}$$

A key question is what kind of bound we should place on this mass for it to be important for the dynamics. A natural constraint arises from holography in the AdS$_2 \times S^2$ region: we should take $m_{\text{eff}}^2 R^2 \leq 1$, so that the dimension of the dual operators is of order one. The resulting bound on the number of modes on the $S^2$ is

$$N \sim l_{\text{max}}^2 \sim \frac{g^2 Q}{32\pi^2 m^2 R^2} = \frac{g^4}{32\pi^3 G m^2 Q} = \frac{g^2 Q_c}{16\pi^2 Q} \,. \tag{27}$$

It is perhaps surprising that this bound is *inversely* proportional to $Q$, whereas the intuition from the Landau levels might have led us to expect a result proportional to $Q$. The reason for this behaviour is that as we increase $Q$, $R$ gets larger, so the curvature of the space goes down and the bound we are imposing on $m_{\text{eff}}^2$ gets tighter.

If we had just considered the bound $k_\perp^2 < B$ discussed around (11) we would have had $N \sim l_{\text{max}}^2 \sim BR^2 \sim Q$. Thus, the bound from requiring that $m_{\text{eff}}^2 R^2 \leq 1$ is stronger so long as $Q > \frac{g}{4\pi} \sqrt{Q_c} \sim 10^{20}$.

At $Q \sim Q_c$, the number of Alfven wave modes this bound would permit is of order one (in fact, smaller than one because of the numerical factors, but the $l = 0$ mode is always allowed as it has $m_{\text{eff}}^2 = 0$). It only becomes large when $Q$ is significantly smaller than $Q_c$; there is still however a large range of possible values of $Q$ where we get large values of $N$. We saw before that $Q \sim Q_c$ corresponded to $R \sim 10^7 m$; if we are interested in wormholes big enough for a human to pass through, so say $R \sim 1m$, this corresponds to $Q \sim 10^{-8} Q_c \sim 10^{33}$, which gives $N \sim 10^4$. The maximum number of "light" Alfven waves is at the cross-over to the bound $k_\perp^2 < B$, where $N \sim Q \sim 10^{20}$.

This looks encouraging. Sadly, we will see that there are stronger restrictions on $m_{\text{eff}}^2$, which lead to smaller values of $N$, restricting us to smaller values of the charge.

## 3.1 Wormhole in a single universe

Our aim is to modify the black hole solution with the quantum stress tensor of the Alfven wave modes to obtain a traversable wormhole geometry. As in [1], our target is a solution with a wormhole with two mouths in a single asymptotically flat spacetime. We will summarize the desired geometry here, following [1], and discuss the extent to which we can obtain it in a construction based on the Alfven wave modes in Section 6.

The geometry is composed of three regions, as shown in Figure 1, with overlapping ranges of validity: a wormhole throat, described by a nearly AdS$_2 \times S^2$ geometry in the global coordinates (23), a throat region at each wormhole mouth, described by the black hole solution (20) (with opposite charges $\pm Q$ at each mouth, as the magnetic field lines flowing into one end of the wormhole flow out at the other) and a flat region with a dipole magnetic field

$$A = \frac{Q}{2}(\cos\theta_1 - \cos\theta_2) d\phi \,, \tag{28}$$

where $\phi$ is the angle around an axis through the two charges, and $\theta_{1,2}$ is the angle between the axis and the line from the point we measure the field at to the plus or minus charge. The two wormhole mouths are separated by a distance $d$ in this approximately flat region; to have a flat space approximation between the wormhole mouths we need $d \gg R$, the separation is larger than the size of the mouths. We assume that the AdS$_2$ throat geometry is valid up to

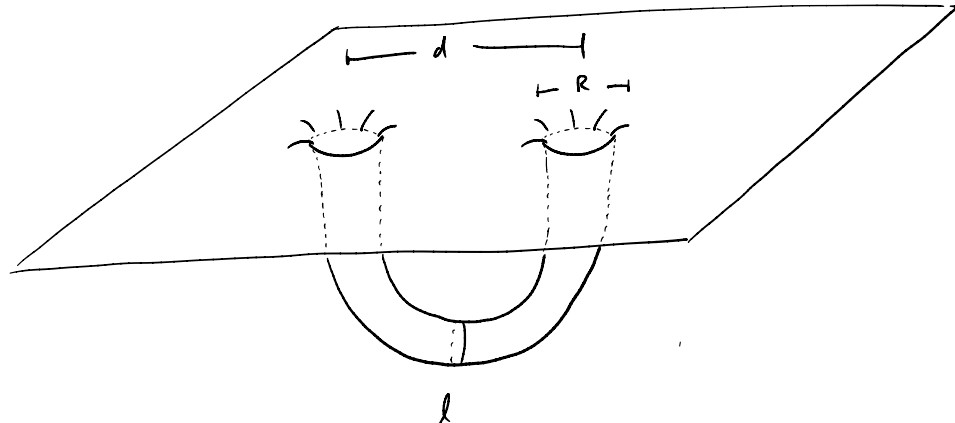

Figure 1: The wormhole geometry: $R$ denotes the radius of the wormhole throat, $d$ denotes the distance between the two wormhole mouths in the ambient space, and $\ell$ is the "length" (or more properly the time taken for a signal to go through) of the wormhole.

some cutoff $\sigma = \pm\sigma_c = \pm(\frac{\pi}{2} - \epsilon)$, where we match to the black hole solution at $r \approx R$. We assume that at this matching point $g_{tt} \approx 1$ in the black hole solution, so the time coordinate $t$ in the black hole and asymptotically flat regimes is $t = \frac{R\tau}{\cos\sigma_c} \approx \frac{R}{\epsilon}\tau = \ell\tau$, where we introduce the parameter $\ell$ to measure the "length" of the wormhole. This is more precisely related to the travel time through the wormhole; if an observer jumps in one mouth at $t = t_0$, they emerge from the other mouth at $t = t_0 + \pi\ell$. We assume that this length of the wormhole is larger than the separation, $\ell \gg d$.

This geometry is not a solution to the equations of motion. The first issue is that the two wormhole mouths would attract each other in the asymptotically flat region, so we would not have a time-independent solution. As in [1], we assume this issue is resolved by having the wormhole mouths slowly orbit a common center or by some other force acting on them, and will not address it further. The second issue, which is the focus of our attention, is that the near-horizon limit of a black hole solution is AdS$_2$ in the Rindler coordinates (22); this describes a solution with an Einstein-Rosen bridge connecting the two wormhole mouths, which is not traversable. To have instead a solution where the asymptotically flat geometry is patched on the AdS$_2$ in global coordinates to obtain a traversable wormhole, we need to add some source of negative energy. Our aim is to obtain this negative energy from the quantum fluctuations of the Alfven wave modes.

The idea of [1] is to consider a field which satisfies the massless wave equation in the two-dimensional space along the field lines of the magnetic field. These field lines form closed loops, threading through the wormhole and then connecting back between the wormhole mouths in the dipole region. The Casimir energy of the two-dimensional fields along these field lines will then provide a source of negative energy. In the regime where $\ell \gg d$, the Casimir energy can be approximately calculated by treating the evolution in the dipole region as essentially trivial. We model this by imposing periodic boundary conditions on the fields in the AdS$_2$ throat region.

The resulting wormhole solution can be determined by extremizing the energy from the perspective of the asymptotic region [1]. The wormhole geometry with a throat of "length"

$\ell$ has the same geometry as a constant-time slice of a non-extremal black hole, with energy above extremality $E = \frac{R^3}{8G_N \ell^2}$.[5] Taking into account the negative Casimir energy from the bulk fields gives a total energy

$$E = \frac{R^3}{8G_N \ell^2} + E_C(\ell). \tag{29}$$

We expect to have a traversable wormhole solution at the value of $\ell$ that extremizes this energy, if $E_C$ is sufficiently negative to make this total energy negative at the extremum. In [1] this analysis was applied to the Landau levels of a massless fermion. This does not help us make large wormholes in our own universe, as there is no exactly massless fermion to work with. We thus want to apply it instead to the Alfven waves.

## 4 Double-trace coupling in AdS$_2$

As a warm-up for considering the effects of the Alfven wave modes in the full asymptotically flat wormhole geometry, let's consider the effects in a pure AdS$_2$ geometry. We will consider introducing a double-trace coupling for the Alfven wave modes in AdS$_2$, following [3, 14].

We consider a JT gravity theory

$$S = \frac{\phi_0}{2}\chi + \frac{1}{2}\int d^2 x \phi(R+2) + \phi_b \int_\partial du K + S_{\text{matter}}, \tag{30}$$

where $\chi$ is the Euler character of the two-dimensional spacetime, and the matter contribution we focus on is the Alfven wave modes, which we describe by a set of $N$ light scalar fields. This theory can be obtained by dimensional reduction of the higher-dimensional theory in the throat region of the black hole. (See [5, 28] for more discussion of the motivation for considering this theory.) The equation of motion of the $\phi$ field sets $R + 2 = 0$, so the geometry is constant negative curvature, implying that the metric is locally AdS$_2$. We consider a geometry with two boundaries, with boundary conditions

$$ds^2_{bdy} = \frac{du}{\epsilon}, \quad \phi_{bdy} = \phi_b = \frac{\phi_r}{\epsilon}, \tag{31}$$

where $\epsilon$ is a cutoff we will take to zero at the end of the calculation. The dynamical mode in the dilaton-gravity sector is a boundary degree of freedom, specifying how the boundaries are embedded in the global AdS$_2$ geometry (23), given by specifying the global coordinates $\tau_L(u)$, $\tau_R(u)$ as functions of the boundary proper time $u$. The action reduces to [5]

$$S = \int du \left[ -\phi_r \left\{ \tan\frac{\tau_L}{2}, u \right\} - \phi_r \left\{ \tan\frac{\tau_R}{2}, u \right\} \right] + S_{\text{matter}}, \tag{32}$$

where the Schwarzian derivative is $\{f(u), u\} = -\frac{1}{2}\left(\frac{f''}{f'}\right)^2 + \left(\frac{f''}{f'}\right)'$. In the absence of the matter contribution, this action has a solution where the boundaries follow hyperbolic trajectories, $\tan\frac{\tau_R}{2} = \tanh u$, $\tan\frac{\tau_R}{2} = -\coth u$, which corresponds to the boundaries lying at constant $\rho$ in the Rindler-like coordinates of (22); that is, this is the near-horizon version of the black hole solution.

We consider adding to this a coupling between the two boundaries. From the holographic perspective, we want to add a double-trace coupling

$$S_{int} = \lambda \sum_{i=1}^{N} \int du O_L^i(u) O_R^i(u), \tag{33}$$

---

[5]This is actually twice the energy of the non-extremal black holes, to account for the two mouths of the wormhole in the asymptotic region.

where $L, R$ denote the two AdS$_2$ boundaries, and $O^i_{L,R}$ are the operators dual to the Alfven wave modes, which have dimension $\Delta = \frac{1}{2} + \frac{1}{2}\sqrt{1 + 4m^2_{\text{eff}}R^2}$ . We are summing over the $N$ modes, where $N$ is bounded by (27). From the bulk perspective, this implies that we introduce a boundary condition for the bulk fields $\phi^i$ which relates the slow fall-off part of the field at one boundary to the fast fall-off part at the other boundary. For a massless scalar, this boundary condition is simply[6]

$$\phi\big|_{\sigma = -\frac{\pi}{2}} = \lambda \partial_\sigma \phi\big|_{\sigma = \frac{\pi}{2}} \qquad \phi\big|_{\sigma = +\frac{\pi}{2}} = -\lambda \partial_\sigma \phi\big|_{\sigma = -\frac{\pi}{2}} . \tag{34}$$

As argued in [14], the effect of this interaction on the dynamics of the boundaries is summarized by the effective action

$$S = \int du \left[ -\phi_r \left\{ \tan \frac{\tau_L}{2}, u \right\} - \phi_r \left\{ \tan \frac{\tau_R}{2}, u \right\} + \frac{\lambda N}{2^{2\Delta}} \left( \frac{\tau'_R(u)\tau'_L(u)}{\cos^2 \frac{\tau_L(u) - \tau_R(u)}{2}} \right)^{2\Delta} \right] . \tag{35}$$

We look for a solution where $\tau_L$, $\tau_R$ are linear functions of $u$, $\tau_L(u) = \tau_R(u) = \tau' u$. This is a solution of the equations of motion from (35) for any $\tau'$; $\tau'$ is then fixed by a global $SL(2)$ constraint. This constraint arises because writing the theory in terms of the effective coordinate locations $\tau_L(u), \tau_R(u)$ of the boundaries in the global coordinates of (23) introduces a redundancy in our description: changes in the values of the coordinate locations of the boundary under global $SL(2)$ transformations do not change the physical solution. We should treat this redundancy in the variables in our effective action as an emergent gauge symmetry. The associated constraint sets the global $SL(2)$ charges of the solution to zero. The non-trivial constraint in the present situation is

$$Q_0 = -2\tau' + 2\Delta \eta \tau'^{2\Delta - 1} = 0, \tag{36}$$

where $\eta$ is a dimensionless version of the coupling $\lambda$, $\eta = \lambda N^{2\Delta - 1}/(2^{2\Delta}\phi_r^{2\Delta - 1})$.

In [14], fermionic operators with $\Delta = \frac{1}{2}$ were considered, corresponding to massless fermions in the AdS$_2$ bulk. Then (36) is easily solved to give $\tau' = \eta$. The analysis is similar for $\Delta < 1$, where $\tau' \propto \eta^{\frac{1}{2(1-\Delta)}}$. But we are interested in massless and massive scalars, corresponding to $\Delta \geq 1$. For $\Delta = 1$, both terms in (36) are linear in $\tau'$, so there is no solution for $\tau'$; instead the vanishing of the charge appears to fix the coupling $\eta$! As discussed in the previous section, the Alfven wave modes are not precisely massless once we take into account corrections to the effective action, so in fact it is more relevant to consider modes in AdS$_2$ with $\Delta > 1$. There is then a solution to (36), but its character has changed; instead of being directly proportional to the coupling $\eta$, $\tau'$ is inversely proportional to $\eta$.

To better understand the physics of this constraint and to clarify the connection to our later discussion for periodic boundary conditions, we would like to re-derive this behaviour from the simple energy extremization argument mentioned in the previous section. For the double-trace boundary conditions, a scalar field on AdS$_2$ will have a Casimir energy linear in $\lambda$ for small $\lambda$, $\mathcal{E}_C \sim \lambda$. Now for operators of dimension $\Delta$, $\lambda$ is a dimensionful coupling, of dimension $2\Delta - 1$. Thus, it is natural to write $\lambda = \eta'/\ell^{2\Delta - 1}$. The Casimir energy defined with respect to the AdS$_2$ time coordinate can then be written as $\mathcal{E}_C = -\frac{R^3 \eta}{G\ell^{2\Delta - 1}}$ (we choose $\eta'(\eta)$ such that the energy takes this form, where the additional factors are introduced for convenience). In the asymptotically flat space, we are interested in the Casimir energy defined with respect

---

[6]The minus sign in the second relation can be understood by requiring that the equation be invariant under the discrete AdS isometry $\sigma \to \pi - \sigma$ that swaps the two boundaries, or by demanding that the resulting boundary value problem be self-adjoint.

to the asymptotically flat coordinate, which is $E_C = \mathcal{E}_c/\ell$. Thus, the total energy as a function of $\ell$ will be

$$E = \frac{R^3}{G\ell^2} - \frac{R^3\eta}{G\ell^{2\Delta}}\,. \tag{37}$$

We want to set

$$\frac{dE}{d\ell} = \frac{R^3}{G\ell^2}\left(-2\frac{1}{\ell} + \frac{2\Delta\eta}{\ell^{2\Delta-1}}\right) = 0\,. \tag{38}$$

The factor in the bracket matches the constraint (36), with $\ell^{-1}$ playing the role of $\tau'$, which is natural, as $\tau'$ represents the scaling between the AdS$_2$ coordinate and the boundary proper time coordinate, while $\ell^{-1}$ is the scaling between the AdS$_2$ time coordinate and the asymptotically flat time coordinate when we paste the AdS$_2$ near-horizon region into the full asymptotically flat solution.

Thus, we see that for the double-trace boundary condition for $\Delta = 1$, the physical problem with the wormhole solution is that we can't balance the Casimir energy contribution against the gravitational contribution, because they scale in the same way with $\ell$. For $\Delta \neq 1$, the extremum is at $\ell = \ell_c$, where $\ell_c^{2(\Delta-1)} = \Delta\eta$. Then

$$\left.\frac{d^2E}{d\ell^2}\right|_{\ell=\ell_c} = -\frac{4R^3(\Delta-1)}{G\ell_c^4}\,, \tag{39}$$

so for $\Delta > 1$, there is an extremum of the energy, but this is now a maximum of the energy rather than a minimum. There is a solution, but it's unstable.

## 5 Casimir for massive scalar field on AdS

In the previous section, we saw that for a double-trace coupling, the physics for bulk scalar fields is quite different from fermions, because of the difference in conformal dimensions. While the latter provides a good energy source to produce a traversable wormhole, massless and massive scalars do not. From the energy extremization perspective, this is because the Casimir energy for the double-trace deformation scales as $1/\ell^{2\Delta}$, and is not relevant enough to balance the gravitational contribution, which dominates in the IR. .

For periodic boundary conditions, one would expect the Casimir energy for massless scalars to scale as $1/\ell$, just as for fermions, so the relevance of this discussion to the case of most interest to us may not yet be very apparent. However, in this section, we will conduct a careful analysis of the Casimir energy for massless and massive scalars on AdS$_2$, and we will find similar features to the discussion above in the massive case.

We consider AdS$_2$ in global coordinates,

$$ds^2 = \frac{R^2}{\cos^2\sigma}(-d\tau^2 + d\sigma^2)\,. \tag{40}$$

The two boundaries are at $\sigma = \pm\frac{\pi}{2}$. Consider a massive scalar field $\phi(t,\sigma)$ with mass $m$. Writing the field as $\phi(t,\sigma) = e^{-i\omega\tau}f(\sigma)$, the wave equation is

$$f''(\sigma) + \left(-\frac{m^2R^2}{\cos^2\sigma} + \omega^2\right)f(\sigma) = 0\,. \tag{41}$$

We cut off the AdS boundary at some scale $\epsilon$, so we have boundaries at $\sigma_+ = \frac{\pi}{2} - \epsilon$ and $\sigma_- = -\frac{\pi}{2} + \epsilon$. The "length" of the wormhole is $\ell = R/\epsilon$.

## 5.1 Massless scalar field on AdS$_2$

Let us first consider the massless case. We will calculate the Casimir energy for periodic boundary conditions by comparing the spectrum for the standard Dirichlet boundary conditions to that for periodic boundary conditions. Dirichlet boundary conditions for a massless scalar impose $f(\frac{\pi}{2}) = f(-\frac{\pi}{2}) = 0$. In this case, the spectrum is $\omega = n \in 1, 2, 3, \cdots$, i.e. all positive integers. For even $\omega$ the solution is

$$f(\sigma) = \sin(n\sigma), \qquad n \in 2, 4, \cdots \tag{42}$$

and for odd $\omega$ the solution is

$$f(\sigma) = \cos(n\sigma), \qquad n \in 1, 3, 5, \cdots \tag{43}$$

Note the putative zero mode with $n = 0$ is a constant and is thus killed by the boundary conditions. Note also that there is no degeneracy; in particular flipping the momentum $n \to -n$ does not result in a linearly independent solution.

Periodic boundary conditions for a massless scalar require

$$f\left(\frac{\pi}{2}\right) = f\left(-\frac{\pi}{2}\right), \qquad f'\left(\frac{\pi}{2}\right) = f'\left(-\frac{\pi}{2}\right). \tag{44}$$

The allowed energies are now $\omega_n = n \in 0, 2, 4, \cdots$ (i.e. only even integers). For all nonzero $n$ there are two linearly independent solutions:

$$f_{n,1}(\sigma) = \cos(n\sigma), \qquad f_{n,2}(\sigma) = \sin(n\sigma). \tag{45}$$

Thus all modes but the zero mode have a 2-fold degeneracy.

It is straightforward to compute the sum over all the zero point energies and compute the Casimir energies. As in [1], we begin by computing these energies on the flat strip defined by $\sigma \in [-\frac{\pi}{2}, +\frac{\pi}{2}]$, and will then conformally map the resulting answers to AdS$_2$. For the Dirichlet case,

$$E = \frac{1}{2} \sum_{n=1}^{\infty} n \exp(-\epsilon n) = \frac{1}{2\epsilon^2} - \frac{1}{24} + \mathcal{O}(\epsilon), \tag{46}$$

where $\epsilon^{-1}$ is a UV cutoff. Here the finite part is the usual Casimir energy for a CFT on a strip of length $\pi$, i.e. $-\frac{c}{24}$ with $c = 1$. For the periodic spectrum

$$E_{\text{periodic}} = \frac{1}{2} \cdot 2 \sum_{n \text{ even}} n \exp(-\epsilon n) = \frac{1}{2\epsilon^2} - \frac{1}{6}. \tag{47}$$

The universal part of this should be compared to the CFT formula $-\frac{c}{12} \frac{2\pi}{L}$ for the energy on a circle of length $L$, where here $L = \pi$. As in [1], when conformally mapped to AdS$_2$, the Dirichlet boundary condition is $SL(2)$ invariant, so the energy vanishes. Thus, we can calculate the Casimir of interest by taking the difference of the periodic and Dirichlet results, which gives

$$\mathcal{E}_{\text{periodic}} = -\frac{1}{8}, \tag{48}$$

where we denote the Casimir energy in the AdS$_2$ frame by $\mathcal{E}$. As before, there is a scaling by $1/\ell$ to relate this to the Casimir energy with respect to the asymptotically flat time coordinate, so we have $E_C = -\frac{1}{8\ell}$, as expected.

## 5.2 Massive scalar field on AdS$_2$

However, the Alfven waves are not exactly massless scalar fields; as we saw earlier in (19), they get a small but non-zero mass from corrections to the effective field theory. Let us therefore consider the computation for a massive scalar field. In the massive case, the solutions of the wave equation are hypergeometric functions. Two linearly independent solutions are

$$f_\pm(\sigma) = \cos^{\alpha_\pm}(\sigma)\,{}_2F_1\left(-\frac{\omega}{2}+\frac{\alpha_\pm}{2},\frac{\omega}{2}+\frac{\alpha_\pm}{2};\alpha_\pm+\frac{1}{2};\cos^2(\sigma)\right), \tag{49}$$

where

$$\alpha_\pm = \frac{1}{2}\pm\sqrt{\frac{1}{4}+m_{\text{eff}}^2 R^2}\,. \tag{50}$$

Note that $\alpha_+ = \Delta$, $\alpha_- = 1-\Delta$ , and $\alpha_+ > \alpha_-$. These solutions are useful for describing the behaviour near the boundaries at $\sigma = \pm\frac{\pi}{2}$; the solutions $f_\pm$ fall off as $\cos^{\alpha_\pm}(\sigma)$, so $f_+$ corresponds to the normalizable mode and $f_-$ corresponds to the non-normalizable mode. However, these solutions are generically not smooth at $\sigma = 0$. Using standard identities [29], the hypergeometric function can be rewritten as

$$\begin{aligned}
&{}_2F_1\left(-\frac{\omega}{2}+\frac{\alpha_\pm}{2},\frac{\omega}{2}+\frac{\alpha_\pm}{2};\alpha_\pm+\frac{1}{2};\cos^2(\sigma)\right)\\
&=\frac{\Gamma(\alpha_\pm+\frac{1}{2})\Gamma(\frac{1}{2})}{\Gamma(\frac{1}{2}+\frac{\alpha_\pm}{2}-\frac{\omega}{2})\Gamma(\frac{1}{2}+\frac{\alpha_\pm}{2}+\frac{\omega}{2})}{}_2F_1\left(-\frac{\omega}{2}+\frac{\alpha_\pm}{2},\frac{\omega}{2}+\frac{\alpha_\pm}{2};\frac{1}{2};\sin^2(\sigma)\right)\\
&\quad+|\sin\sigma|\frac{\Gamma(\alpha_\pm+\frac{1}{2})\Gamma(-\frac{1}{2})}{\Gamma(\frac{\alpha_\pm}{2}-\frac{\omega}{2})\Gamma(\frac{\alpha_\pm}{2}+\frac{\omega}{2})}{}_2F_1\left(-\frac{\omega}{2}+\frac{\alpha_\pm}{2}+\frac{1}{2},\frac{\omega}{2}+\frac{\alpha_\pm}{2}+\frac{1}{2};\frac{3}{2};\sin^2(\sigma)\right),
\end{aligned} \tag{51}$$

which is continuous but not smooth at $\sigma = 0$ because of the second term.

We obtain the Casimir energy by comparing the spectrum for Dirichlet and periodic boundary conditions. By Dirichlet, we mean that we want a solution which has only the $f_+$ falloff near the boundary. If we consider the solution $f_+(\sigma)$ given above, we then need to choose the energy $\omega$ such that $f_+(\sigma)$ is smooth at $\sigma = 0$. This can be achieved by taking

$$\omega = \alpha_+ + 2r\,, \quad r = 0,1,2,\ldots\,, \tag{52}$$

so that the Gamma function $\Gamma(\frac{\alpha_+}{2}-\frac{\omega}{2})$ in the denominator in the second term in (51) has a pole, and we have just the first term, which is smooth at $\sigma = 0$. In the massless limit $m \to 0$, this reproduces the part of the spectrum with odd $n = 2r + 1$ and even solutions $f(\sigma) = \cos(n\sigma)$, corresponding to the fact that $f_+$ is an even function. To get the other half of the spectrum, we need to consider a solution $f_+(\sigma)\text{sgn}(\sigma)$ (note that this satisfies the equation of motion, and like $f_+(\sigma)$, it is generically not smooth at $\sigma = 0$). This is an odd function, and the solution is smooth if

$$\omega = \alpha_+ + 2r + 1\,, \quad r = 0,1,2,\ldots\,, \tag{53}$$

so that the Gamma function $\Gamma(\frac{1}{2}+\frac{\alpha_+}{2}-\frac{\omega}{2})$ in the denominator in the first term in (51) has a pole, and we have just the second term, which is smooth at $\sigma = 0$ when multiplied by $\text{sgn}(\sigma)$. In the massless limit, this reproduces the part of the spectrum with even $n = 2r + 2$ and odd solutions $f(\sigma) = \sin(n\sigma)$. The Dirichlet spectrum for the massive field thus consists of (52,53).

Now consider periodic boundary conditions,

$$f(\sigma_+) = f(\sigma_-)\,, \qquad f'(\sigma_+) = f'(\sigma_-)\,, \tag{54}$$

with some cutoff $\epsilon$, such that the two edges are at $\sigma_+ = \frac{\pi}{2}-\epsilon$ or $\sigma_- = -\frac{\pi}{2}+\epsilon$. Here we will proceed by constructing solutions which are smooth at $\sigma = 0$, and then imposing the

boundary conditions. The analysis above encourages us to consider separately the even and odd functions of $\sigma$. An even function is obtained in general by considering

$$f(\sigma) = C_+ f_+(\sigma) + C_- f_-(\sigma). \tag{55}$$

Smoothness at $\sigma = 0$ then requires that the coefficient of the second term in (51) vanishes, that is

$$C_+ \frac{\Gamma(\alpha_+ + \frac{1}{2})\Gamma(-\frac{1}{2})}{\Gamma(\frac{\alpha_+}{2} - \frac{\omega}{2})\Gamma(\frac{\alpha_+}{2} + \frac{\omega}{2})} + C_- \frac{\Gamma(\alpha_- + \frac{1}{2})\Gamma(-\frac{1}{2})}{\Gamma(\frac{\alpha_-}{2} - \frac{\omega}{2})\Gamma(\frac{\alpha_-}{2} + \frac{\omega}{2})} = 0. \tag{56}$$

For even functions, we automatically have $f(\sigma_+) = f(\sigma_-)$, and the non-trivial boundary condition is $f'(\sigma_+) = f'(\sigma_-) = 0$. For small $\epsilon$,

$$f'(\sigma_+) \approx \alpha_+ C_+ \epsilon^{\alpha_+} + \alpha_- C_- \epsilon^{\alpha_-} = 0. \tag{57}$$

Combining these two equations, we have a condition for $\omega$,

$$\frac{\Gamma(\alpha_+ + \frac{1}{2})}{\Gamma(\frac{\alpha_+}{2} - \frac{\omega}{2})\Gamma(\frac{\alpha_+}{2} + \frac{\omega}{2})} - \frac{\alpha_+}{\alpha_-} \epsilon^{2\Delta-1} \frac{\Gamma(\alpha_- + \frac{1}{2})}{\Gamma(\frac{\alpha_-}{2} - \frac{\omega}{2})\Gamma(\frac{\alpha_-}{2} + \frac{\omega}{2})} = 0, \qquad \text{even sector} \tag{58}$$

where the power of $\epsilon$ is $\alpha_+ - \alpha_- = 2\Delta - 1$. Similarly, odd functions are

$$f(\sigma) = (C_+ f_+(\sigma) + C_- f_-(\sigma))\text{sgn}(\sigma). \tag{59}$$

Smoothness at $\sigma = 0$ then requires that the coefficient of the first term in (51) vanishes, that is

$$C_+ \frac{\Gamma(\alpha_+ + \frac{1}{2})\Gamma(\frac{1}{2})}{\Gamma(\frac{1}{2} + \frac{\alpha_+}{2} - \frac{\omega}{2})\Gamma(\frac{1}{2} + \frac{\alpha_+}{2} + \frac{\omega}{2})} + C_- \frac{\Gamma(\alpha_- + \frac{1}{2})\Gamma(\frac{1}{2})}{\Gamma(\frac{1}{2} + \frac{\alpha_-}{2} - \frac{\omega}{2})\Gamma(\frac{1}{2} + \frac{\alpha_-}{2} + \frac{\omega}{2})} = 0. \tag{60}$$

For odd functions, the non-trivial boundary condition is $f(\sigma_+) = f(\sigma_-) = 0$. For small $\epsilon$,

$$f(\sigma_+) \approx C_+ \epsilon^{\alpha_+} + C_- \epsilon^{\alpha_-} = 0. \tag{61}$$

Combining these two equations, we have a condition for $\omega$,

$$\frac{\Gamma(\alpha_+ + \frac{1}{2})}{\Gamma(\frac{1}{2} + \frac{\alpha_+}{2} - \frac{\omega}{2})\Gamma(\frac{1}{2} + \frac{\alpha_+}{2} + \frac{\omega}{2})} - \epsilon^{2\Delta-1} \frac{\Gamma(\alpha_- + \frac{1}{2})}{\Gamma(\frac{1}{2} + \frac{\alpha_-}{2} - \frac{\omega}{2})\Gamma(\frac{1}{2} + \frac{\alpha_-}{2} + \frac{\omega}{2})} = 0. \qquad \text{odd sector} \tag{62}$$

Taking $\epsilon \to 0$ at fixed $m_{\text{eff}}$, (57,61) set $C_- = 0$ in both the even and odd sectors, and the periodic case reduces to the Dirichlet case. Physically, this happens because the non-normalizable mode blows up near the boundary, so to satisfy a periodic boundary condition in the limit as we extend our box to the boundary of the spacetime we must set the coefficient of the non-normalizable mode to zero. However, if we take the mass $m \to 0$ at fixed $\epsilon$, the factor of $\alpha_-$ in (57) goes to zero as we take $m \to 0$, and we must set $C_+$ to zero for even modes. That is consistent with the analysis of the massless case in the previous section, where we found that the periodic spectrum included odd functions which vanished at the boundary and even functions which were finite at the boundary.

Thus, at fixed small $m_{\text{eff}}$, $\epsilon$, the odd part of the periodic spectrum is always close to the Dirichlet spectrum (as in the massless case, where the odd parts of the periodic and Dirichlet spectra coincided), but whether the even part is close to the Dirichlet spectrum or the massless periodic spectrum depends on the order of limits with which we take $\epsilon \to 0$ or $m_{\text{eff}} \to 0$, or equivalently the size of the coefficient $\frac{\alpha_+}{\alpha_-} \epsilon^{2\Delta-1}$ in (58). To see this explicitly, let's substitute $\omega = \alpha_+ + 2r + 1 + \delta\omega^{\text{odd}}$ in (62). If $\epsilon$ is small, we may expand in $\delta\omega^{\text{odd}}$ to find to first order:

$$\delta\omega^{\text{odd}} \approx -2\epsilon^{2\Delta-1} \frac{(-1)^r \Gamma(\alpha_- + \frac{1}{2})\Gamma(1 + \alpha_+ + r)}{r!\,\Gamma(\alpha_+ + \frac{1}{2})\Gamma(-\frac{1}{2}(\alpha_+ - \alpha_-) - r)\Gamma(\frac{3}{2} + r)}, \tag{63}$$

so the spectrum indeed agrees with the Dirichlet one up to corrections vanishing as $\epsilon^{2\Delta-1}$.

For the even case, if $\frac{\alpha_+}{\alpha_-}\epsilon^{2\Delta-1}$ is small, then setting $\omega = \alpha_+ + 2r + \delta\omega^{\mathrm{even}}$ and expanding the gamma function on the first term in (58) gives

$$\delta\omega^{\mathrm{even}} \approx -2\frac{\alpha_+}{\alpha_-}\epsilon^{2\Delta-1}\frac{(-1)^r\Gamma(\alpha_-+\frac{1}{2})\Gamma(\alpha_++r)}{r!\Gamma(\alpha_++\frac{1}{2})\Gamma(-\frac{1}{2}(\alpha_+-\alpha_-)-r)\Gamma(\frac{1}{2}+r)}, \tag{64}$$

a small correction to the Dirichlet spectrum. Alternatively, if $\frac{\alpha_+}{\alpha_-}\epsilon^{2\Delta-1}$ is large, for $r \neq 0$ we can set $\omega = \alpha_- + 2r + \delta\omega^{\mathrm{even}'}$ and expand the gamma function in the second term of (58) , to give

$$\delta\omega^{\mathrm{even}'} \approx -2\frac{\alpha_-}{\alpha_+}\epsilon^{1-2\Delta}\frac{(-1)^r\Gamma(\alpha_++\frac{1}{2})\Gamma(\alpha_-+r)}{r!\Gamma(\alpha_-+\frac{1}{2})\Gamma(\frac{1}{2}(\alpha_+-\alpha_-)-r)\Gamma(\frac{1}{2}+r)}, \tag{65}$$

a small correction to the massless periodic spectrum.[7]

Thus, for massive fields of fixed mass, at small $\epsilon$ (which is large $\ell = R/\epsilon$), the spectrum for periodic boundary conditions differs from Dirichlet by an amount that scales as $\delta\omega \sim \epsilon^{2\Delta-1} \sim 1/\ell^{2\Delta-1}$. As explained above, we measure the Casimir energy for the periodic boundary condition relative to that with the Dirichlet boundary condition; thus we will find an effective Casimir energy only when the spectra differ, i.e. it will be of order: $\mathcal{E}_{\mathrm{periodic}} \sim 1/\ell^{2\Delta-1}$. Hence the Casimir energy with respect to the asymptotically flat time coordinate is $E_C \sim 1/\ell^{2\Delta}$. Interestingly, this gives the same behaviour as for the double-trace boundary condition we considered in the previous section. Massless fields are a special case, as the factor of $\alpha_-$ vanishes, so the spectrum remains different from the Dirichlet one in the limit as $\epsilon \to 0$, giving the finite answer exhibited in (48).

In the situation we are interested in, we have a large number of fields with a range of masses, and we consider some large but finite value of $\ell$. The fields then split roughly into two groups: for masses small enough that $\frac{\alpha_-}{\alpha_+}\epsilon^{1-2\Delta}$ is small, we have essentially the massless spectrum, and a finite Casimir $\mathcal{E}_C$ approximately independent of $\ell$, while for larger masses we have the situation above with $\mathcal{E} \sim 1/\ell^{2\Delta-1}$. The crossover is when the $m_{\mathrm{eff}}^2 \to 0$ limit of $\frac{\alpha_-}{\alpha_+}\epsilon^{1-2\Delta}$ is of $\mathcal{O}(1)$; using $\epsilon = \frac{\ell}{R}$, this happens when:

$$m_{\mathrm{eff}}^2 \sim m_c^2 = \frac{1}{\ell R}. \tag{66}$$

For long wormholes, this is a much smaller mass than the simple holographic bound we had previously, $m_{\mathrm{eff}}^2 < \frac{1}{R^2}$, and it will lead to an $\ell$-dependent restriction on the number of modes contributing non-trivially to the Casimir energy.

This slightly lengthy computation simply shows that for sufficiently long length scales $\ell$ a very small mass makes a large difference, and the calculation shows that the relevant mass scale is the geometric mean of $\ell$ and $R$.

---

[7]For $r = 0$, the analysis is slightly more subtle; the size of $\delta\omega^{\mathrm{even}'}$ is still controlled by $\frac{\alpha_+}{\alpha_-}\epsilon^{2\Delta-1}$, but the power is different. Setting $\omega = \alpha_- + \delta\omega^{\mathrm{even}'}$ in (58), we have approximately

$$\frac{\Gamma(\alpha_++\frac{1}{2})}{\Gamma(\frac{\alpha_+}{2})^2} - \frac{\alpha_+}{\alpha_-}\epsilon^{2\Delta-1}\frac{\Gamma(\alpha_-+\frac{1}{2})}{\Gamma(-\frac{\delta\omega^{\mathrm{even}'}}{2})\Gamma(\alpha_-+\frac{\delta\omega^{\mathrm{even}'}}{2})} = 0.$$

From the analysis for $r \neq 0$, we guess that $\delta\omega^{\mathrm{even}'}$ is large compared to $\alpha_-$; then we should neglect $\alpha_-$ relative to $\delta\omega^{\mathrm{even}'}$ in the second gamma function, to get $(\delta\omega^{\mathrm{even}'})^2 \approx -\frac{1}{2\pi}\alpha_-\epsilon^{1-2\Delta} \approx \frac{1}{2\pi}m^2R^2\epsilon^{1-2\Delta}$. This is large compared to $\alpha_-^2$ so the approximation is consistent. For our argument, this power is not important; what matters is that the spectrum is close to Dirichlet for small $\frac{\alpha_+}{\alpha_-}\epsilon^{2\Delta-1}$, and close to the massless for large $\frac{\alpha_+}{\alpha_-}\epsilon^{2\Delta-1}$.

# 6 Wormhole with periodic boundary conditions

In the full wormhole geometry, assuming an FFE description is valid in both the $AdS_2$ and the dipole regions, the Alfven waves behave as two-dimensional fields propagating along the field lines of the magnetic field. These field lines form closed loops, threading through the wormhole and then connecting back between the wormhole mouths in the dipole region. If the length $\ell$ of the wormhole in the $AdS_2$ region is long compared to the separation $d$ in the dipole regione, we can think of this as simply giving a periodic boundary condition for the fields in the $AdS_2$ region.

In the previous section, we saw that massive fields with such a periodic boundary condition give a non-trivial Casimir energy only for $m_{\text{eff}}^2 < \frac{1}{\ell R}$. This implies a strong restriction on the number of modes for which we have an appreciable Casimir energy. Using (26), we have:

$$N \sim l_{\text{max}}^2 = \frac{g^2 Q}{32\pi^2 m^2} \frac{1}{\ell R} = \frac{g^3}{32\pi^{5/2}\sqrt{G} m^2} \frac{1}{\ell}. \tag{67}$$

Note that this is independent of $Q$; the number of modes that satisfy this bound does not scale at all with the flux. More importantly, it also goes down as we make the wormhole longer. Thus, even though individual fields have a Casimir energy which scales as $1/\ell$, the total Casimir energy from these fields in the asymptotically flat frame is

$$E_C = -\frac{N}{8\ell} \sim -\frac{1}{\ell^2}, \tag{68}$$

showing the same scaling as in the double-trace analysis for massless scalars. The further contributions from Alfven wave modes with $m_{\text{eff}}^2 > \frac{1}{\ell R}$ give a Casimir energy scaling as $1/\ell^{2\Delta}$, so they are negligible at large $\ell$ relative to the contributions we keep here.

Thus, we have the same problem as in the double-trace case: the total energy that we want to extremise is

$$E = \frac{R^3}{G\ell^2} - \frac{g^3}{256\pi^{5/2} m^2 l_p \ell^2}, \tag{69}$$

and both terms have the same scaling with $\ell$. Thus, we have not succeeded in stabilizing a wormhole at large values of $\ell$.

Fortunately, this is not the end of the story; there is a difference between this case and the double-trace case, which is that the $1/\ell^2$ scaling of the Casimir here came from counting the number of modes with small enough mass. Now $N \sim 1/\ell$ only if $m_{\text{eff}}^2 < \frac{1}{\ell R}$ is the strongest bound on the effective mass of the Alfven wave modes. But if we can choose the parameters so that the energy in (69) is negative, the wormhole will reduce its energy by becoming shorter, and the bound $m_{\text{eff}}^2 < \frac{1}{\ell R}$ becomes weaker at smaller $\ell$, so we can expect that eventually some other physics could take over and stabilise the wormhole.

Indeed, in our analysis above, we treated the dipole regime as simply imposing a periodic boundary condition, identifying the fields at the two wormhole mouths. This will be a good approximation if the field does not vary significantly across the dipole regime, propagating over a distance of order $d$ in flat space. This requires $m_{\text{eff}}^2 \ll 1/d^2$. If $\ell$ becomes sufficiently small, this could become a stronger bound on $N$.[8] Let's suppose $\ell$ does become sufficiently small that this dipole bound matters. Then

$$E = \frac{R^3}{G\ell^2} - \frac{g^2 Q}{256\pi^2 m^2 d^2 \ell}. \tag{70}$$

---

[8]Note that this bound is also always stronger than the simple bound $m_{\text{eff}}^2 < \frac{1}{R^2}$, as we assume $d > R$. The number of Alfven wave modes allowed by the dipole bound is linear in $Q$, $N = \frac{g^2 Q}{32\pi^2 m^2 d^2}$, so this bound will also be stronger than the restriction $N < Q$ so long as the separation $d$ is bigger than the Compton wavelength of the electron. Also note that the crossover to the dipole bound can happen in the regime where $\ell \gg d$, if $d$ is sufficiently large.

This has a minimum at

$$\ell_{\min} = \frac{512\pi^{7/2}m^2 d^2 l_p Q^2}{g^5} \, . \tag{71}$$

This is in the regime where the dipole bound matters if $\ell_{\min}R < d^2$, which implies

$$Q^3 < \frac{g^6}{512\pi^4 m^2 G} \, . \tag{72}$$

For such values of $Q$, we have $E < 0$ in (69). Thus, there is a self-consistent story: if $Q$ is small enough, $E$ is given by (69) at large $\ell$, and is negative. The dynamics then drives $\ell$ to decrease, increasing the number of Alfven wave modes contributing to the Casimir. At sufficiently small $\ell$ this increase is cut off by the requirement that $m_{\text{eff}}^2 < 1/d^2$, and the energy is given by (70). This has a minimum at $\ell_{\min}$, giving a stable traversable wormhole solution, supported by the negative Casimir energy of the Alfven wave modes.

Unfortunately, the values of $Q$ satisfying (72) are pretty small: $Q < 10^{12}$, which correspond to a wormhole of size $R < 10^{-21}$m, or about $(1000 \text{ TeV})^{-1}$. This mechanism is thus only possible for very small wormholes indeed. Even though we have argued that FFE is a good description for much larger values of $R$, the tight bound on the effective mass means that only a small number of Alfven wave modes contribute to the Casimir energy, and we need to be at small $Q$ for the second term in (69) to overcome the first term[9]. We have thus not succeeded in building significantly larger wormholes than in [1]. Furthermore, at such short scales, our analysis, which is based ultimately on considering QED+Maxwell, will break down disastrously, not only because FFE is clearly not a good description, but because of the contribution of other standard model and possibly beyond-the-standard-model fields; we need to take into account the effects discussed in [2].

Our simplest attempt at using these collective modes to stabilize the wormhole was not successful; nevertheless it would be very interesting indeed to see if some variant thereof could allow the existence of a large stable traversable wormhole using the low-energy field content of our universe. Our results hinge delicately on the structure of higher-derivative corrections to FFE, which to our knowledge are not very well-understood. Indeed if the Alfven wave density of states differs significantly from the (conservative) assumptions that led to (19), our conclusions may be modified. Interesting directions for future work are to explore further such higher-derivative corrections, and in a more general sense the relation between the FFE description and the underlying QED+Maxwell theory[10], particularly in the regime of very strong fields relevant to the small values of the charge we consider above. It would also be interesting to explore other effects of the Alfven wave modes and more generally the strong magnetic fields in black holes in the FFE regime.

## Acknowledgements

This work was supported in part by STFC through grant ST/P000371/1. We are grateful to S. Gralla, D. Hofman and N. Poovuttikul for discussions and collaboration on related issues.

---

[9]Demanding that the field $B \sim \frac{Q}{d^2}$ in the dipole region be strong enough for FFE to be valid, and using $Q \sim 10^{12}$ from above, we further find that $d < 10^5 \, m^{-1}$, with $m$ the mass of the electron. Though there is a large hierarchy between $d$ and $R$, this is still rather small in everyday terms.

[10]See [25] for a recent investigation of the validity of FFE using holography.

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
