# Peer review of "Towards traversable wormholes from force-free plasmas"

_SciPost Physics, doi:SciPost Phys. 12, 086 (2022)_

## Round 1 · Referee Report · Anonymous · 2021-5-1

Report

The paper studies the possibility of making travesable wormholes in 4d Einstein gravity by using Casimir energy of Alfeven waves of force-free electrodynamics(FFE). Although the idea of using Casimir energy is not new, and ultimately authors encountered serious obstacles to realizing this idea, FFE near black holes is an interesting topic which recently drew some attention in the literature. Also taking into account that the paper is clearly written and authors discuss various physical effects, I recommend the paper
for publication with major revisions.
Here is the list of questions I would like the authors to address:
1) FFE seems to be an emergent phenomena when standard electrodynamics interacts with matter. The authors considered the correction to the (total) stress-energy tensor from Alfven waves. Does this take into account the stress-energy tensor of underlying matter?
2) It seems that eventually authors impose simple periodic boundary conditions on matter. Is Section 4 necessary then?
3) The conclusion of the paper seems that it is hard to sustain a wormhole with FFE in 4d. In higher dimensions the black hole story is well studied(AdS2 near horizon, 2d Casimir energy, etc), _except_ the dispersion relation for Alfven waves. It would be nice if the author could comment on this.
4) The lengthy calculation in Section 5.2 results in the condition $m^2 < 1/(R l)$. The authors explain this result as: "...AdS kinematics imply that the relevant mass scale is the geometric mean...". I think it would be very helpful for readers to see this kinematic estimate explicitly. Naively, in eq. (41) mass is responsible for $1/(\cos^2 \sigma)$ potential, which is cutoff at $|\sigma-\pi/2| \sim R/l $. Since $\omega \gtrsim 1$, the effect of the mass is negligible for $m l \lesssim 1$.

---

## Round 1 · Referee Report · Anonymous · 2021-5-18

Report

Constructing traversable wormholes in the low energy theory of our universe is an important and exciting problem. This paper proposes an interesting new idea to achieve this by using collective light modes arising from the force-free electrodynamics description (which has proven useful in astrophysical setups). At the end, the simple setup described in the paper doesn't lead to a consistent wormhole solution. Nonetheless, the idea is interesting and deserves further study. The nearly-AdS2 perspective on these collective modes is also of some interest. The analysis is clear and convincing. Therefore, I recommend this paper for publication in SciPost Physics.

That said, I would like to ask the authors some minor clarifications/questions:

1) Is it be possible to clarify a little bit the physical setup, to give some intuition to a reader who doesn't know much about FFE? If I understand correctly, the Alfven modes are supposed to describe coherent excitations of the plasma and EM field, so does this mean that the hypothetical wormhole must be filled with plasma? Does this has any physical significance?

2) The Alfven wave modes are supposed to follow the magnetic field lines in closed loops threading the wormhole. A question that I haven't seen addressed is the following: is it clear that the FFE approximation is also valid in the outside region, far from the black holes? Shouldn't this put a constraint on the outside distance d?

3) In [1], a large magnetic field was crucial to have a large number of Casimir modes. This doesn't seem to be the case here where it's only needed for the FFE to be valid. So could we repeat the analysis in a more realistic astrophysical setup? For example by replacing the RN black hole with near-extreme Kerr in a strong magnetic field?

---

## Round 1 · Referee Report · Anonymous · 2021-5-21

Strengths

1) Very well written paper
2) Exceptionally clear
2) Very novel idea
3) Clear description of calculations

Weaknesses

1) The mechanism they study unfortunately doesn't lead to large traversable wormholes

Report

This paper attempts to construct large traversable wormholes using standard physics arising from charged particles interacting with a strong magnetic field, in regimes where a particular low energy effective theory is useful. The problem of constructing large wormholes without invoking physics beyond the standard model is of great interest, and the approach taken here is a novel and interesting mechanism, certainly worthy of exploration.

This is a very well written paper, that is exceptionally clear and well organized -- and overall very enjoyable to read. I therefore recommend the paper for publication.

---

## Round 2 · Referee Report · Anonymous (Referee 1) · 2021-6-27

Report

Dear Authors,
Thank you to your clarifications. The only remaining question I have is about Casimir energy in AdS2. More specifically, eq. (66).
Parameter $\epsilon$ was first introduced right below eq. (41). So I believe you have a typo in the definition of $\epsilon$ above eq. (44).
Unless I am very much mistaken, in the limit $\Delta \rightarrow 1, m R \rightarrow 0$ and with $\epsilon=R/l$, the quantity $\alpha_- \epsilon^{1-2\Delta}/\alpha_+$ goes to $m l$. This implies that in eq. (66) it should be $1/l^2$ instead of $1/(l R)$. The same conclusion can be reached in a very simple way, by looking at eq. (41) and requiring that the $m/\cos^2$ potential is negligible.
  • validity: -
  • significance: -
  • originality: -
  • clarity: -
  • formatting: -
  • grammar: -

Author:  Nabil Iqbal  on 2021-07-09  [id 1562]

(in reply to Report 1 on 2021-06-27)
Category:
reply to objection

Dear referee,

In response to your question below:

Thank you to your clarifications. The only remaining question I have is about Casimir energy in AdS2. More specifically, eq. (66). Parameter ϵ was first introduced right below eq. (41). So I believe you have a typo in the definition of ϵ above eq. (44).

We don’t quite understand this comment; in particular, there is no definition of \epsilon above eq. (44).

Unless I am very much mistaken, in the limit Δ →1, mR → 0 and with ϵ = R/l, the quantity alpha_- epsilon^(1- 2 Delta)/alpha_+ goes to ml

We actually disagree with this statement; in particular, from Eq. (50), in the limit of small m, alpha_- goes like m^2 R^2, so alpha_- epsilon^(1- 2 Delta)/alpha_+ = alpha_-/epsilon = m^2 R l, and not ml. (We find it possible the referee made an incorrect Taylor expansion of Eq. (50), thinking erroneously that \alpha_- \sim (m R).)

This implies that in eq. (66) it should be 1/l^2 instead of 1/(lR). The same conclusion can be reached in a very simple way, by looking at eq. (41) and requiring that the m/cos^2 potential is negligible.

Based on the above argument, eq (66) is correct as written. As far as we can see there is no extremely simple argument to establish this scaling; obtaining it requires our explicit calculation. The referee's proposed bound would be obtained by evaluating the potential at the boundary — i.e. looking at the potential at \sigma = \frac{\pi}{2} - \epsilon — however this gives a result which is too strong, just as evaluating the potential at a generic point in the interior results in a bound m^2 < 1/R^2, which is too weak.

Anonymous on 2021-09-13  [id 1758]

(in reply to Nabil Iqbal on 2021-07-09 [id 1562])

Dear Authors,
My concern is that in eq. (65) for $r=0$ mode the Gamma function $\Gamma(\alpha_- + r)$ actually behaves as
$1/\alpha_-$ which cancels overall $\alpha_-$ so the correction is not small anymore. It becomes of order $\epsilon^{-1}$.

Anonymous on 2021-08-09  [id 1647]

(in reply to Nabil Iqbal on 2021-07-09 [id 1562])

The referee is correct that there is a typo above eq (66), where epsilon should be R/l, not l/R.

However, in their other comment, they appear to have confused the mode number r with R, the curvature radius of the AdS2. (At small m, \alpha_{-} scales like mR, not mr). Our claim is that the relevant expansion parameter is m^2 R l. As far as we can see this is equally valid for the case with r=0.

Anonymous on 2021-07-16  [id 1574]

(in reply to Nabil Iqbal on 2021-07-09 [id 1562])
Category:
reply to objection

Dear Authors, thank you for the clarifications.

there is no definition of $\epsilon$ above eq. (44).

I am sorry, I meant $\epsilon$ above eq. (66).

I agree with your reasoning regarding $\alpha_\pm$. However, I still have some doubts regarding the Casimir energy computation. My intuition is that for finding when the mass is irrelevant it is not necessary to find the exact spectrum, simply identifying what is the relevant expansion parameter is enough. If $m^2 r l$ is the right expansion parameter, contrary to naive $m l$ then this is a separate interesting result. Especially since there have been lots of discussions regarding wormholes recently.

To wit, you have found an explicit correction to energies, eq. (65)(also I believe it should be $\omega=\alpha_- + 2r+ \delta \omega$ above it). For all non-zero $r$ this correction is indeed small if $m^2 r l$ is small. However, for $r=0$, there is dangerous $\Gamma(\alpha_-) \approx 1/\alpha_- = 1/mr$. So the correction to $\omega=0$ is not necessarily small. I presume this correction should be found from the exact quantization condition (58). If this correction turns out to be irrelevant for Casimir energy then it will clearly demonstrate that $m^2 r l$ is the right parameter.

---

## Round 2 · Referee Report · Anonymous (Referee 2) · 2021-6-30

Report

The authors have satisfactorily answered my comments.

---

## Round 2 · Author Response

We thank the referees for their careful reading and useful feedback. We are happy that they overall find the paper informative and suitable for publication. We have addressed their concerns, as detailed below.

To Referee 1:

1) FFE seems to be an emergent phenomena when standard electrodynamics interacts with matter. The authors considered the correction to the (total) stress-energy tensor from Alfven waves. Does this take into account the stress-energy tensor of underlying matter?

So, FFE as conventionally formulated assumes that the matter stress energy tensor can be neglected, as that of the electromagnetic field is considered to be higher. The correction we compute in a specific model is a slight deformation away from that limit, so indeed takes into account some of the stress-energy of that matter. We thank the referee for the question, and have clarified this with two comments:

— on p3, we have added the line "In its conventional formulation, it is usually understood that the stress-energy of the electromagnetic field is much higher than that of the charged matter screening the electric field, which can thus be neglected.”

— on p7, we added "Similarly, the stress energy of the Alfven wave can be understood as that of an approximately massless collective scalar field moving in the AdS2 directions; microscopically however this stress energy comes both from the electromagnetic degrees of freedom and from the fermion degrees of freedom bosonized into the field Φ.”

2) It seems that eventually authors impose simple periodic boundary conditions on matter. Is Section 4 necessary then?

We feel that Section 4 is helpful and would like to keep it, as otherwise it may appear that a slight modification of the periodic boundary conditions would be sufficient to evade the later conclusions; in fact it seems to us it is a generic feature of the construction that this is difficult to realize.

3) The conclusion of the paper seems that it is hard to sustain a wormhole with FFE in 4d. In higher dimensions the black hole story is well studied(AdS2 near horizon, 2d Casimir energy, etc), except the dispersion relation for Alfven waves. It would be nice if the author could comment on this.

Actually, in d> 4 FFE is likely to have a very different structure indeed. The lack of phenomenological considerations mean that the case of d > 4 has not been studied by the plasma physics community and remains to be developed. In a more general context, a key role is played by the 1-form symmetry associated with magnetic flux conservation (as emphasized in Ref 21); but in general d this is a (d-3)-form symmetry, and the structure of the theory will be quite different.

Said in a slightly different manner, in d = 5 the magnetically charged black hole must be a black string and not a black hole, and thus things seem different. We feel that this is outside the scope of this paper (which was attempting to address the possibility of creating these wormholes in our own 4d universe).

4) The lengthy calculation in Section 5.2 results in the condition m^2 < 1/(RL). The authors explain this result as: "...AdS kinematics imply that the relevant mass scale is the geometric mean...". I think it would be very helpful for readers to see this kinematic estimate explicitly. Naively, in eq. (41) mass is responsible for .... potential, which is cutoff at ... Since ...the effect of the mass is negligible for ....

[some equations deleted above for formatting reasons; the original comment can be seen in the first submissions page]

We actually do not think there is a direct kinematic argument that shoes that the geometric mean is the relevant scale; rather it appears to follow from a somewhat detailed computation. We agree the original wording was somewhat misleading, and have rephrased the line to read on p16:

"This slightly lengthy computation simply shows that for sufficiently long length scales l a very small mass makes a large difference, and the calculation shows that the relevant mass scale is the geometric mean of l and R.”

To Referee 2:

1) Is it be possible to clarify a little bit the physical setup, to give some intuition to a reader who doesn't know much about FFE? If I understand correctly, the Alfven modes are supposed to describe coherent excitations of the plasma and EM field, so does this mean that the hypothetical wormhole must be filled with plasma? Does this has any physical significance?

As FFE is a rather large subject, we feel that we can’t really do justice to it here and would like to refer readers to the excellent reviews on the subject, e.g. Ref [17]. Regarding the interaction with conventional plasma physics, we have already addressed this on p3 with the line:

"The theory is often considered in situations with a plasma density, but the theory is still useful for describing fluctuations around vacuum electromagnetic backgrounds satisfying the degeneracy condition F ∧ F = 0, if the background magnetic field is strong enough; in response to fluctuations, charges can be easily pair- produced, screening the electric field to zero over long distance scales. We will consider the theory in this setting.”

and further on p8 as

"The black hole is a solution of Einstein gravity coupled to a Maxwell field, but the magnetically charged black holes also satisfy the FFE equations of motion, and can be thought of as solutions of FFE coupled to gravity. As mentioned in the previous section, FFE is usually thought of as a theory of plasmas, but it includes as solutions any degenerate vacuum Maxwell field, and an FFE description is useful if the field is strong enough that fluctuations about this background that would produce electric fields violating the FFE equation are efficiently screened by charges produced by vacuum fluctuations; in this case we expect the low-energy fluctuations to be collective plasma modes (such as Alfven waves) rather than free photon excitations.”

and thus would prefer to not add any further discussion. (We note that FFE is likely valid in other physical situations as well, and in fact its precise domain of validity does not seem to be something that is completely well understood in the current plasma physics literature).

2) The Alfven wave modes are supposed to follow the magnetic field lines in closed loops threading the wormhole. A question that I haven't seen addressed is the following: is it clear that the FFE approximation is also valid in the outside region, far from the black holes? Shouldn't this put a constraint on the outside distance d?

We thank the referee for this good point, which we have now addressed in footnote 8 on p18. The referee is correct that there is a bound on d as well; it is much larger than R (the radius of the throat), but still rather small in everyday terms, which we have now explicitly computed.

3) In [1], a large magnetic field was crucial to have a large number of Casimir modes. This doesn't seem to be the case here where it's only needed for the FFE to be valid. So could we repeat the analysis in a more realistic astrophysical setup? For example by replacing the RN black hole with near-extreme Kerr in a strong magnetic field?

Though the meaning of “large” may be somewhat subjective, we actually still think that the field required is still rather large. i.e. in the approximation we work in, FFE is valid only for B greater than the critical field defined in (15), which is roughly speaking related to having a field large enough that it is reasonable to pair-produce charges from the vacuum. We note that there may have been some confusion here from a typo above Eq (15) where an inequality was accidentally reversed (which we have now corrected to read B > B_{\star}, in agreement with footnote 3). Thus we think it remains a bit unlikely astrophysically.

Finally, Referee 3 did not ask for any further corrections.

Again, we thank all the referees for their careful reading and insightful questions, which we feel have improved the paper.

---

## Round 2 · List of Changes

The point-by-point list is available above in our responses to the referee comments.

---

## Round 3 · Referee Report · Anonymous · 2021-10-22

Report
Dear Authors,
Thank you for answering my questions. The result about the Casimir energy in AdS is very important and can have a big significance for the field of wormhole study in general. This is why I want to make sure it is indeed determined by $m^2 R l$ as you claim.
Now I completely agree with your results for the even spectrum of periodic BC.
However, I noticed that that the Dirichlet spectrum, eqns. (63), (64) have a problem: for large $r$, the corrections $\delta \omega$ behave as $r^{1-2 \alpha_-}$, so it grows faster than the original, linear in $r$, answer. So the approximation breaks down. How does it affect the final answer for the Casimir energy?
Also it would be useful to check if the same happens with the odd spectrum for periodic BC.
I appreciate your patience.
Author: Nabil Iqbal on 2021-12-05 [id 2010]
(in reply to Report 1 on 2021-10-22)Dear referee,
We thank you for the detailed reading.
Actually, the large-r growth of $\delta\omega$ seen in Eq (63) is not physical, and is an artifact of assuming that the frequency shift $\delta\omega$ is *small*; for a fixed value of $\epsilon$, this ceases to be a good assumption for sufficiently large $r$, and thus one should not linearize the Gamma functions in Eq (62) when finding the frequency shift.
There is of course no issue with numerically solving the original defining equation Eq. (62) numerically. We have done this below for the first 100 zeros with ($\epsilon^{2\Delta-1} = 0.01$,$ m^2R^2 = 0.05$). These parameters have been picked to show both that the two expressions agree for small $r$, (where the domain of validity increases as $\epsilon$ is made smaller), but also that for large $r$ the growth of the *exact* answer saturates to be much slower than linear and there is no breakdown of the Casimir energy calculation. (See Figure in file "pic.pdf", which should be attached to this comment).
We stress that this has no impact on our answers, which do not require the precise $r$-dependence in (63) and really use only the overall scaling with $\epsilon$ in Eq (63). As $\epsilon$ goes to zero, the approximation used to obtain (63) remains valid for larger and larger $r$, and the leading non-analytic dependence on \epsilon indeed arises from Eq (63).
As this is all rather technical (and the considerations are only important if one takes the particular order of limits where $\epsilon$ is held fixed, UV cutoff on r is taken to infinity), we do not think that this requires a revision in the paper.
Attachment:
pic.pdf
Anonymous on 2022-02-06 [id 2162]
(in reply to Nabil Iqbal on 2021-12-05 [id 2010])The fact that the correction (63) grows with $r$ stems from a simple fact that the authors expanded the hypergeometric function in eq. (61). Taking $m=0$ in eq. (62) produces $\sin(\pi \omega/2)- \epsilon \omega \cos(\pi \omega/2)=0$ instead of $\sin((\pi/2-\epsilon) \omega)=0$. For large $\omega$ the difference becomes important. Using the full hypergeometric function, one can easily check numerically that the proper set of eigenenergies is $\omega = (2r + 1 + \alpha_+)/(1-2 \epsilon/\pi)$ instead of $(2r + 1 + \alpha_+)$.
So I agree that it is enough to require $m^2 R l$ to be small, as the Authors have claimed.
One simple way to derive this requirement is to apply standard quantum-mechanical first-order perturbation theory to the eigenproblem (41), treating $m^2 R^2/\cos(\sigma)^2$ as a perturbation(previously I suggested comparing this term to $1$, but this is obviously too crude).
I do not have any further questions and I recommend the paper for publication.
Anonymous on 2022-01-23 [id 2118]
(in reply to Nabil Iqbal on 2021-12-05 [id 2010])Dear Authors,
Thank you for a detailed reply.
Your graph indicates that for large $r$ the correction becomes finite even for small $\epsilon$. I did the same calculation for smaller $\epsilon$ and got a similar result(please see the attached plot). It means that for large $r$ the approximation does break down: the shift $\delta \omega$ becomes a constant(my numerical experiments suggest this constant is 1)
This shifts Casimir energy by an order 1 amount, as the sum $\sum_{r=0}^\infty r$ and $\sum_{r=0}^{\infty}(r+c)$ are different. So I disagree with your claim that it does not affect your result: Casimir energy is sensitive to precise $r$ dependence.
Attachment:
ads_periodic_correction.pdf

---

## Round 3 · Author Response

List of changes
Before equations (63) and (66) we correct the mode-numbering of the frequencies in the inline formula for \omega.
We add footnote 7 on page 6 to explain a potential subtlety in the lowest mode.

You are currently on this page

---

## Round 3 · List of Changes

Before equations (63) and (66) we correct the mode-numbering of the frequencies in the inline formula for \omega.
We add footnote 7 on page 6 to explain a potential subtlety in the lowest mode.

You are currently on this page

---

## Editorial Decision

published